# Association between Paraoxonase/Arylesterase Activity of Serum PON-1 Enzyme and Rheumatoid Arthritis: A Systematic Review and Meta-Analysis

**DOI:** 10.3390/antiox11122317

**Published:** 2022-11-23

**Authors:** Gian Luca Erre, Stefania Bassu, Roberta Giordo, Arduino A. Mangoni, Ciriaco Carru, Gianfranco Pintus, Angelo Zinellu

**Affiliations:** 1Dipartimento di Medicina, Chirurgia e Farmacia, Università degli Studi di Sassari, Viale San Pietro, 8, 07100 Sassari, Italy; 2Dipartimento di Scienze Biomediche, Università degli Studi di Sassari, 07100 Sassari, Italy; 3College of Medicine, Mohammed Bin Rashid University of Medicine and Health Sciences, Dubai P.O. Box 505055, United Arab Emirates; 4Discipline of Clinical Pharmacology, College of Medicine and Public Health, Flinders University, Sturt Road, Bedford Park, SA 5042, Australia; 5Department of Clinical Pharmacology, Flinders Medical Centre, Southern Adelaide Local Health Network, Flinders Drive, Bedford Park, SA 5042, Australia

**Keywords:** oxidative stress, rheumatic diseases, biomarkers, paraoxonase activity, arylesterase activity

## Abstract

Background: A decrease in serum paraoxonase (PON-1) and arylesterase (ARE) activity has been reported in rheumatoid arthritis (RA) patients and linked to chronic inflammation and impaired antioxidant defense. Methods: A systematic review and meta-analysis were performed to critically appraise the current evidence on plasma/serum concentrations of PON-1 and ARE activity in RA patients and healthy controls. The Web of Science, PubMed, Scopus, and Google Scholar databases were searched from inception to November 2021. We used random-effects meta-analysis. The risk of bias was estimated using the Joanna Briggs Institute Critical Appraisal Checklist tool. The certainty of the evidence was assessed with GRADE. The study complied with the PRISMA statements and was registered in PROSPERO (CRD42022345380). Results: Seventeen studies reported PON-1 activity (1144 RA patients, 797 controls) and ten reported ARE activity (1367 RA patients, 1037 controls). RA patients had significantly lower PON-1 (SMD = −1.32, 95% CI −1.94 to −0.70; *p* < 0.001) and ARE activity (SMD = −0.91, 95% CI −1.37 to −0.46; *p* < 0.001). There was substantial heterogeneity (PON, I2 97%; ARE, 95.7%, *p* < 0.001 for both). There was no publication bias. The pooled SMD values did not significantly change after sensitivity analysis. The certainty of the evidence was very low due to the observational nature of the studies and the large heterogeneity. Conclusion: Our meta-analysis has shown that both serum PON-1 and ARE activity are significantly lower in RA patients, suggesting a deficit in antioxidant defense mechanisms in this disease.

## 1. Introduction

Rheumatoid arthritis (RA) is a systemic autoimmune disease characterized by chronic inflammation, preferentially involving the peripheral joints [1]. However, dysregulated immune response and exuberant inflammation are implicated in the development of systemic features of RA, including cardiovascular disease [2,3]. Pathogenic pathways leading to cardiovascular disease in RA populations are multiple, including micro and macrovascular endothelial dysfunction, accelerated arterial stiffening with modifications of central hemodynamics, and early atherosclerosis [4,5,6,7]. Consequently, RA patients are at increased risk of atherosclerotic coronary artery disease, atrial fibrillation, stroke, and arrhythmogenic sudden death [2,3,4,8]. Of note, currently available cardiovascular risk scores, based on conventional cardiovascular risk factors, usually underestimate the risk of future cardiovascular events in the RA population [2]. 

This suggests that other processes, such as systemic inflammatory burden, metabolic derangement, and increased oxidative stress, are involved in the development of structural and functional cardiovascular alterations in RA as well as in other systemic autoimmune diseases [9,10,11,12]. In particular, the reduction of anti-atherogenic activity of HDL-cholesterol in the context of oxidative stress and systemic inflammation may have a role in the promotion of cardiovascular disease in RA. 

One of the anti-atherogenic effects of HLD-cholesterol has been linked to paraoxonase-1 (PON-1), a calcium-dependent HDL-associated esterase with three hydrolytic activities, PON-1 activity, arylesterase (ARE) activity, and lactonase activity [13]. PON-1 was shown to protect HDL and LDL from oxidation [14] and to slow the process of atherosclerotic plaque formation by inhibiting the differentiation of monocytes into macrophages and foam cells [15]. Accordingly, an association between impaired PON-1 activity and increased cardiovascular risk [16] has been reported in a prospective large cohort study of patients undergoing coronary angiography [17].

Of note, conflicting data are available regarding PON-1 and ARE activities in RA patients until now. Therefore, we sought to summarize the available evidence by conducting a systematic review and meta-analysis of studies comparing PON-1 and ARE activity between RA patients and the general population.

## 2. Materials and Methods

### 2.1. Search Strategy, Eligibility Criteria, and Study Selection

We performed a systematic search of papers in the bibliographic databases Web of Science, PubMed, Google Scholar, and Scopus, from inception to November 2021, using the following key terms: “Paraoxonase” or “PON” or “Paraoxonase-1” or “PON-1” or “arylesterase” and “rheumatoid arthritis”. Two investigators independently screened the abstracts to establish relevance and reviewed the full articles. The following criteria of eligibility were used to select relevant papers: (i) measurement of PON-1 and/or ARE activity in serum or plasma; (ii) case-control studies comparing RA and controls; (iii) sample size of study ≥10; (iv) papers written in English; and (v) full-text accessible. 

Additional studies were also retrieved by searching the reference lists of selected papers. A third investigator was involved when needed to resolve disagreements between the first two reviewers. The risk of bias of selected publications was evaluated using the Joanna Briggs Institute (JBI) Critical Appraisal Checklist, with a score of ≥5, 4, and <4 indicating low, moderate, and high risk of bias, respectively [18]. The Grades of Recommendation, Assessment, Development, and Evaluation (GRADE) Working Group system was used to measure the strength of the certainty of the evidence. The GRADE system included the following items: (a) the study design: observational vs. randomized; (b) the risk of bias evaluated using the JBI tool; (c) the presence of unexplained heterogeneity; (d) the indirectness; d) the imprecision of results measured according to study sample size, 95% confidence interval width, and threshold crossing; (e) the effect size (SMD < 0.5, SMD 0.5–0.8, SMD > 0.8, indicating small, moderate, and large effect sizes, respectively [19]); (f) the presence of publication bias [20,21]. 

The study adhered to the Preferred Reporting Items for Systematic Reviews and Meta-Analyses (PRISMA) 2020 statement [22]. The protocol was registered in the International Prospective Register of Systematic Reviews (PROSPERO, CRD42022345380).

### 2.2. Statistical Analysis

We calculated standardized mean differences (SMDs) and 95% confidence intervals (CIs) and used forest plots to show differences in PON-1 and ARE concentrations between RA patients and controls. A *p*-value < 0.05 was considered statistically significant. When only median and IQR values (or ranges) were available, the mean and standard deviation values were calculated as recommended by Hozo et al. [23]. Graph Data Extractor software was used when needed to extrapolate numeric data from graphs. We used the Q statistic, with a significance level of *p* < 0.10, to measure SMD heterogeneity across studies.

As suggested in the Cochrane handbook [24], we considered I^2^ < 30% as no or slight heterogeneity and I^2^ ≥ 30% as moderate or substantial heterogeneity. We used a random effect model meta-analysis (inverse-variance) in cases of moderate or substantial heterogeneity meta-analysis [25]. 

We conducted a sensitivity analysis, excluding individual studies sequentially, in order to explore the relative influence on the effect size of each study. The Begg’s adjusted rank correlation test and the Egger’s regression asymmetry test were used to assess publication bias [26,27]. Moreover, the Duval and Tweedie “trim and fill” test was used to evaluate, and eventually amend, publication bias [28]. We conducted univariate meta-regression analyses to assess the presence of a significant association between the estimated effect size and the following parameters: age, proportion of males, continent of study conduction, and year of publication.

Analyses were conducted using Stata 17 (STATA Corp., College Station, TX, USA). 

## 3. Results

### 3.1. Systematic Research

We initially identified 732 studies. We excluded 707 studies as they were irrelevant or duplicates, and five because of missing data (one), duplicate data (two), or different study design (not a case-control study, two). Thus, we included 20 studies in the meta-analysis (Figure 1) [29,30,31,32,33,34,35,36,37,38,39,40,41,42,43,44,45,46,47,48].

The characteristics of studies included in the meta-analyses are summarized in Table 1. 

Serum PON-1 values were described in 17 studies [29,30,31,32,34,35,37,38,39,40,41,42,43,45,46,47,48] (Figure 2), while ARE serum activity was reported in ten [31,32,33,34,35,36,38,40,44,47] (Figure 3).

### 3.2. Meta-Analysis of PON-1 Activity

#### 3.2.1. Study Characteristics

Seventeen studies with a total of 1144 RA patients (mean age 49 years, 25% males) and 797 healthy controls (mean age 49 years, 31% males) were assessed [29,30,31,32,34,35,37,38,39,40,41,42,43,45,46,47,48]. Ten studies were performed in Asia [29,30,31,32,34,35,37,38,43,48], five in Europe [39,41,42,46,47], one in America [40], and one in Africa [45].

#### 3.2.2. Risk of Bias

The risk of bias was estimated to be low in 10 studies [30,34,35,37,39,40,41,45,46,47], moderate in six [29,31,32,38,43,48], and high in one [42] (Appendix A).

#### 3.2.3. Results of Individual Studies and Syntheses

PON-1 activity in RA and controls in the seventeen retrieved studies is reported in the forest plot in Figure 2. 

PON-1 activity compared to controls was higher in one study (difference 0.56) [34] and lower in the remaining 16 studies (mean difference range, −0.12 to −6.10). However, in six of these 16 studies, the difference in PON-1 activity between RA and controls did not reach the statistical significance [38,39,40,41,43,48]. 

Substantial heterogeneity between studies was observed (I^2^ = 97.0%, *p* < 0.001). In the random-effects model meta-analysis, PON-1 activity was found to be significantly lower in RA patients compared to controls (SMD = −1.32, 95% CI −1.94 to −0.70; *p* < 0.001). 

The effect size remained substantially unaltered in the sensitivity analysis performed omitting in turn each study (effect size ranged between −1.43 and −1.01, Figure 4). Of note, also after excluding the four largest studies (accounting for ~50% of the overall sample size), PON-1 activity persisted to be significantly lower in RA compared to controls (SMD = −1.22, 95% CI −1.83 to −1.60, *p*  =  0.001; I^2^  =  93.9%, *p*  <  0.001). 

In addition, after omitting two studies that significantly influenced the funnel plot symmetry [35,37] (Figure 5), the estimated SMD, although attenuated, remained significant (SMD = −0.83, 95% CI −1.24 to −0.41, *p* < 0.001). Accordingly, an extreme heterogeneity between studies was still measured (I^2^ = 92.9%, *p* < 0.001).

#### 3.2.4. Publication Bias

We found no significant publication bias on the remaining 15 studies (Begg’s test, *p* = 0.09; Egger’s tests, *p* = 0.14). Accordingly, the “trim-and-fill” procedure did not add any missing studies to the funnel plot (Figure 6). 

#### 3.2.5. Meta-Regression and Sub-Group Analysis

We found no significant association between the effect size and age (t = 0.92, *p* = 0.37), proportion of males (t = −0.25, *p* = 0.81), publication year (t = 0.17, *p* = 0.87), or sample size (t = −0.73, *p* = 0.47) in univariate meta-regression analysis. 

Sub-group analysis detected larger, albeit not significantly different (*p* = 0.56), between-group differences in serum PON-1 activity in Asian (SMD = −1.58; 95% CI −2.69 to −0.48, *p* = 0.005; I^2^ = 97.7%, *p* < 0.001) vs. European studies (SMD = −0.98; 95% CI −1.86 to −0.11, *p* = 0.027; I^2^ = 96.6%, *p* < 0.001) (Figure 7).

#### 3.2.6. Certainty of Evidence 

The quality of the evidence was downgraded to “very low”, considering the observational design of the studies (−2) and the high heterogeneity (−1) (Appendix A). 

### 3.3. Meta-Analysis of ARE Activity

#### 3.3.1. Study Characteristics

Ten studies investigating a total of 1367 RA patients (mean age 48 years, 22% males) and 1037 controls (mean age 47 years, 30% males) were evaluated [31,32,33,34,35,36,38,40,44,47]. Six studies were performed in Asia [31,32,34,35,36,38], two in America [33,40], and two in Europe [44,47].

#### 3.3.2. Risk of Bias

The risk of bias was low in six studies [33,34,35,36,40,47] and moderate in four [31,32,38,44] (Appendix A). 

#### 3.3.3. Results of Individual Studies and Syntheses

The forest plot for ARE activity in patients with RA and controls in the ten retrieved studies is reported in Figure 3. 

RA patients had significantly higher ARE activity than controls (difference of 0.65) in one study [34] and lower ARE activity than controls in the remaining nine studies (mean difference range, −0.05 to −4.35). In two of these nine studies, the difference between RA and controls in ARE activity was not statistically significant [40,44]. 

A random-effects model meta-analysis was conducted due to the presence of substantial heterogeneity between studies (I^2^ = 95.7%, *p* < 0.001). Pooled results showed a significantly lower ARE activity in RA patients compared to controls (SMD = −0.91, 95% CI −1.37 to −0.46; *p* < 0.001). The pooled SMD direction remained stable after sensitivity analysis (effect size ranged between −1.54 and −0.61 (Figure 8).

The between-group difference in ARE activity was relatively larger (SMD = −1.01, 95% CI −1.77 to −0.43, *p* = 0.001; I^2^ = 93.8%, *p* < 0.001) after omitting the two largest studies (~70% of the whole sample size). 

The estimated size effect remained significant (SMD = −0.61, 95% CI −0.98 to −0.24, *p* < 0.001) also after excluding the study [33], which influences the funnel plot graph symmetry (Figure 9). However, the heterogeneity between studies remained extremely high (I^2^ = 93.5%, *p* < 0.001).

#### 3.3.4. Publication Bias

There was no significant publication bias in the remaining 15 studies (Begg’s test, *p* = 0.46; Egger’s test, *p* = 0.31). As expected, the “trim-and-fill” procedure did not detect missing studies (Figure 10). 

#### 3.3.5. Meta-Regression and Sub-Group Analysis

We found no significant associations between the effect size and age (t = −1.22, *p* = 0.26), proportion of males (t = 0.25, *p* = 0.81), publication year (t = 0.22, *p* = 0.84), or sample size (t = 0.69, *p* = 0.51) in the univariate meta-regression analysis. In sub-group analysis, significant between-group differences in serum ARE activity were observed in Asian (SMD = −0.62; 95% CI −1.01 to −0.22, *p* = 0.002; I^2^ = 85.7%, *p* < 0.001) but not American (SMD = −2.34; 95% CI −6.25 to 1.57, *p* = 0.24; I^2^ = 98.6%, *p* < 0.001) or European studies (SMD = −0.78; 95% CI −2.23 to 0.67, *p* = 0.29; I^2^ = 98.2%, *p* < 0.001) (Figure 11).

#### 3.3.6. Certainty of Evidence 

We downgraded the certainty of the evidence to “very low” due to the observational design of the studies (−2) and the high heterogeneity (−1) observed (Appendix A). 

## 4. Discussion

PON-1 is a 43 kDa calcium-dependent glycoprotein with 355 amino acid residues [49,50]. It is in the liver and released into the circulation, where it is associated mainly with HDLs and, to a lesser extent, very low-density lipoproteins and chylomicrons [51]. HDL-bound PON-1 shows higher enzymatic activity than free PON-1. PON-1 is then transferred from the liver to several tissues [52], where it binds to cell membranes, protecting lipids against peroxidation [51]. In addition, PON1 protects low-density lipoprotein (LDL) from oxidation and have important anti-inflammatory effects [53]. Another important role of PON-1 is to protect against exposure of specific organophosphates (OPs) by arylesterase activity. Studies on knockout mice have shown that PON-1 provides a significant protection against exposure to the OPs, chlorpyrifos oxon and diazoxon [54]. Furthermore, PON-1 shows lactonase activity that allows it to hydrolyze homocysteine thiolactone (HCTL). HCTL is a toxic metabolite that reacts with protein lysines, thus leading protein inactivation and dysfunction. It is amply described as increased HCTL serum concentrations being associated with an increased risk of developing cardiovascular, neurological, and autoimmune diseases [55,56,57,58]. However, the physiological relevance of PON-1 regarding HCTL is uncertain due to its low specific activity [59]. Therefore, PON-1 exhibits a broad range of enzymatic activities towards various types of substrates (lactonase, thiolactonase, arylesterase, and aryldialkylphosphatase activities) in a single active site [60].

The pooled SMD values observed in our meta-analysis indicate a significant reduction in PON-1 and ARE activity in RA patients compared to controls. This suggests that specific pathways related to RA pathogenesis may have an impact on the levels of antioxidants, including PON-1 and ARE. 

In one study [34], PON-1 and ARE activity were higher in RA than controls. The reason behind this discrepancy is not clear, but differences in the laboratory methodology and in the specific characteristics of the population enrolled in the study (e.g., inflammatory state, cholesterol profile, and drugs) may have affected these results. 

Chronic inflammation in RA has been shown to affect cholesterol lipoprotein structure and functionality, potentially affecting HDL and PON-1 activity. However, apart from inflammation, other largely unexplored mechanisms, such as genetic polymorphisms and epigenetics, may at least partially influence differences between groups. 

Of note, sensitivity analysis performed to explore the nature of the extreme and unexplained between-study heterogeneity observed in the PON-1 and ARE activity meta-analyses did not significantly change the results. Moreover, in the meta-regression analysis, we found no significant association between effect size and age, gender, sample size, year of publication, and the geographic areas of publication. Differences in disease activity and the use and dose of different anti-inflammatory and autoimmune medications may have, at least in part, affected heterogeneity between studies. 

Finally, the overall certainty of the evidence was very low due to the specific observational design and the large heterogeneity between studies included in the metanalyses.

## 5. Conclusions

In this systematic review and metanalysis, we showed significantly reduced PON-1 and ARE activity in RA patients when compared to controls. This supports the concept that systemic inflammation and the autoimmune response are closely linked to the substantial impairment of antioxidant defense mechanisms in RA. Whether this phenomenon translates into the observed increase of atherosclerotic burden and cardiovascular disease in RA should be explored in future, large, prospective population studies. 

## Figures and Tables

**Figure 1 antioxidants-11-02317-f001:**
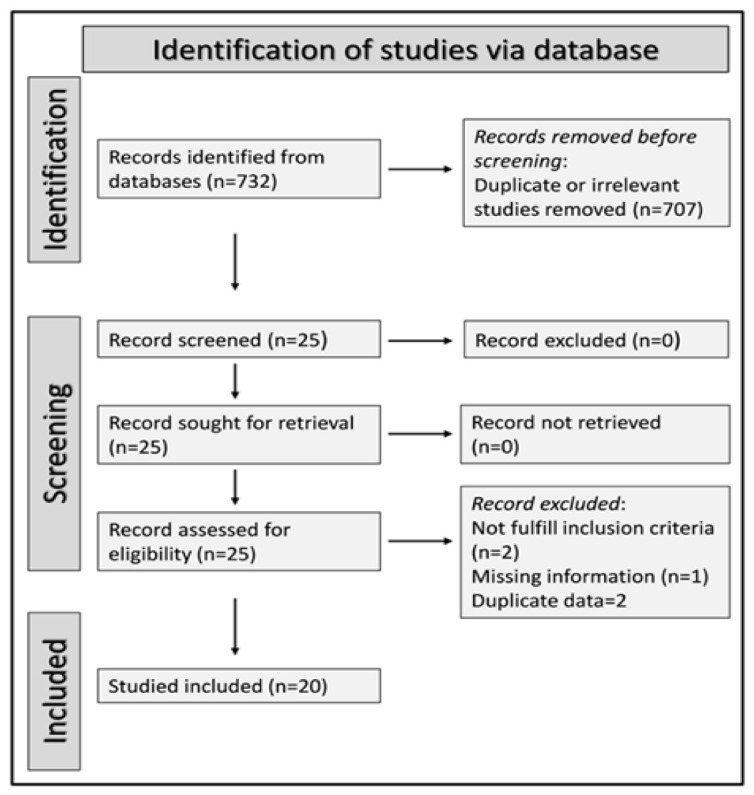
Flow chart describing the screening process.

**Figure 2 antioxidants-11-02317-f002:**
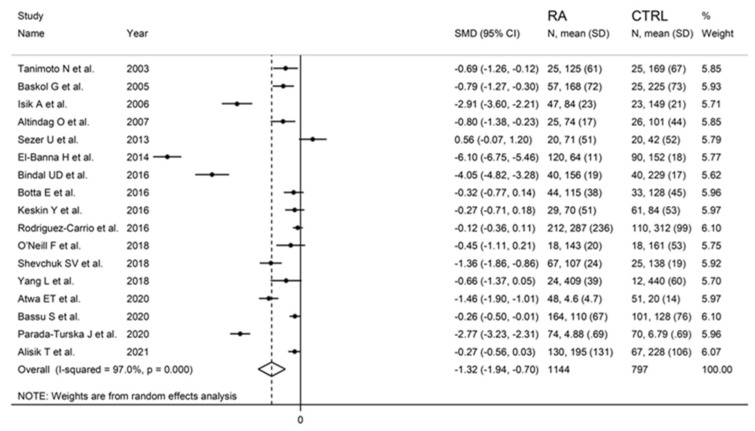
Forest plot of studies examining serum PON-1 values of RA and controls.

**Figure 3 antioxidants-11-02317-f003:**
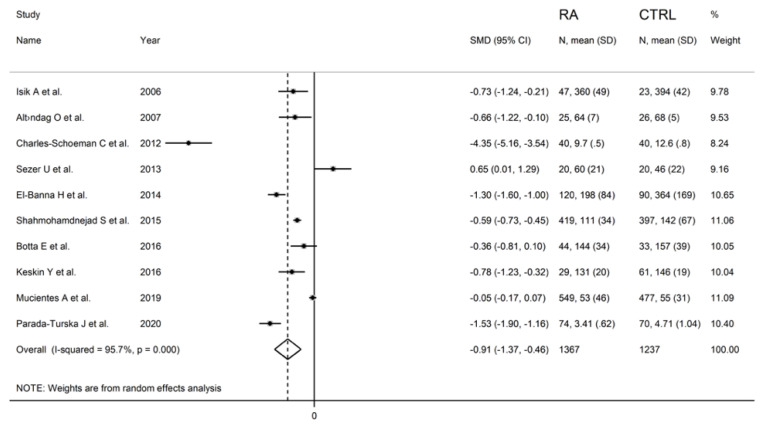
Forest plot of studies examining serum ARE values of RA and controls.

**Figure 4 antioxidants-11-02317-f004:**
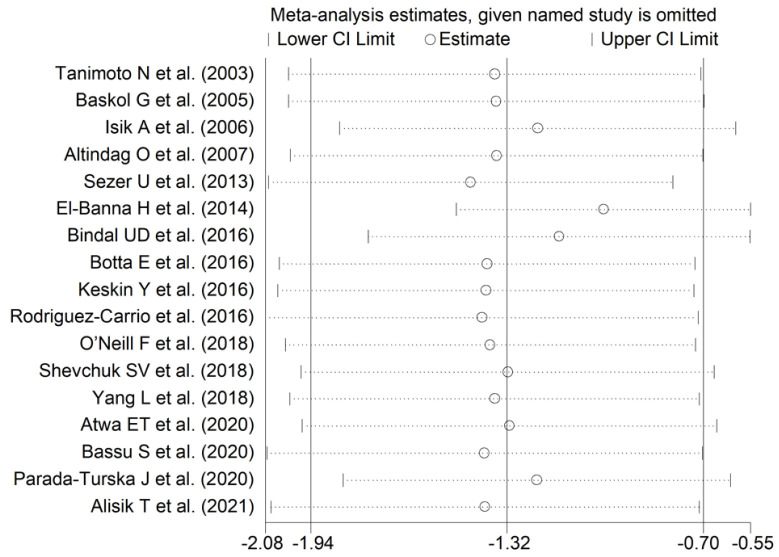
Sensitivity analysis of the association between serum PON-1 values and RA. The effect size (hollow circles) for each study represents the effect size of the meta-analysis performed excluding that study.

**Figure 5 antioxidants-11-02317-f005:**
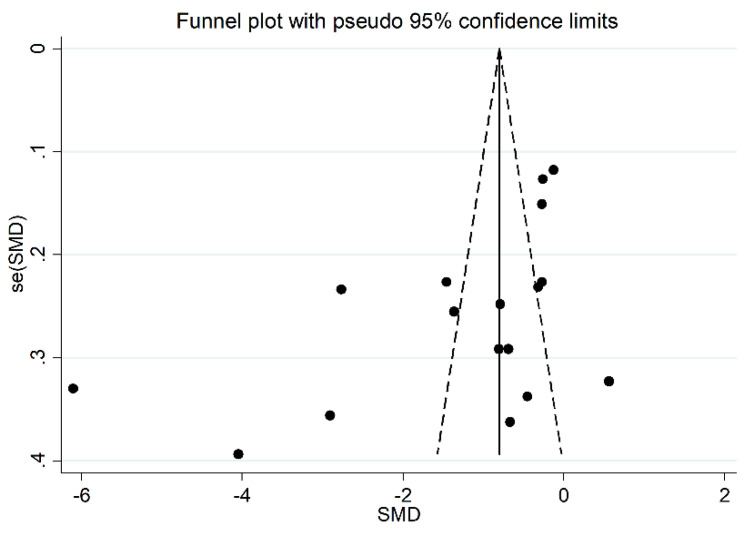
Funnel plot of the 17 studies included in the meta-analysis reporting the association between serum PON-1 activity and RA.

**Figure 6 antioxidants-11-02317-f006:**
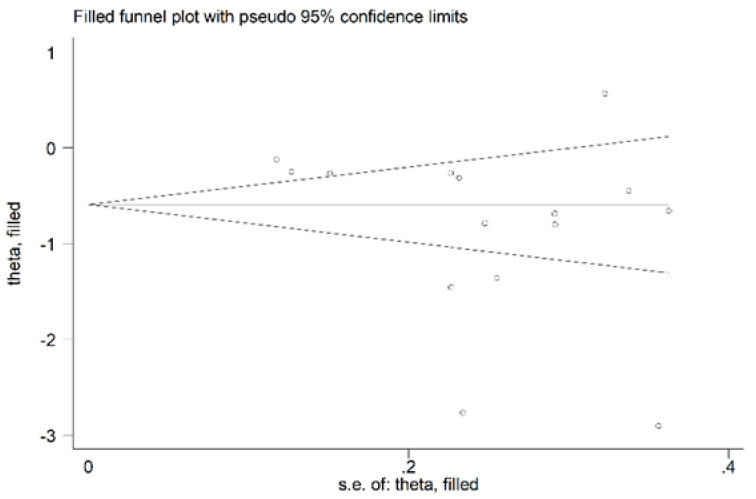
Funnel plot of studies reporting associations between serum PON-1 values and RA after trimming and filling. Dummy studies and genuine studies are represented by enclosed circles and free circles, respectively.

**Figure 7 antioxidants-11-02317-f007:**
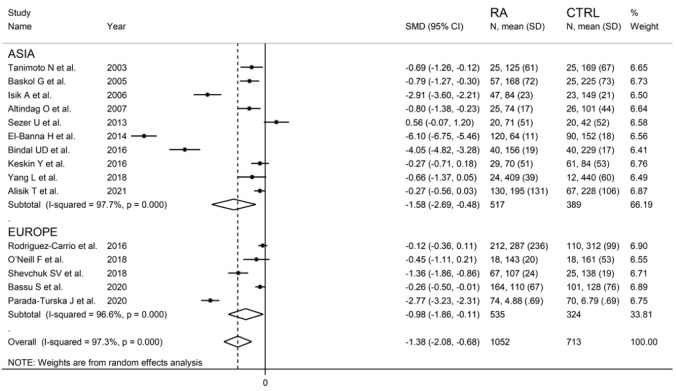
Forest plot of studies examining serum PON-1 concentration in RA and controls according to continent where the study was conducted.

**Figure 8 antioxidants-11-02317-f008:**
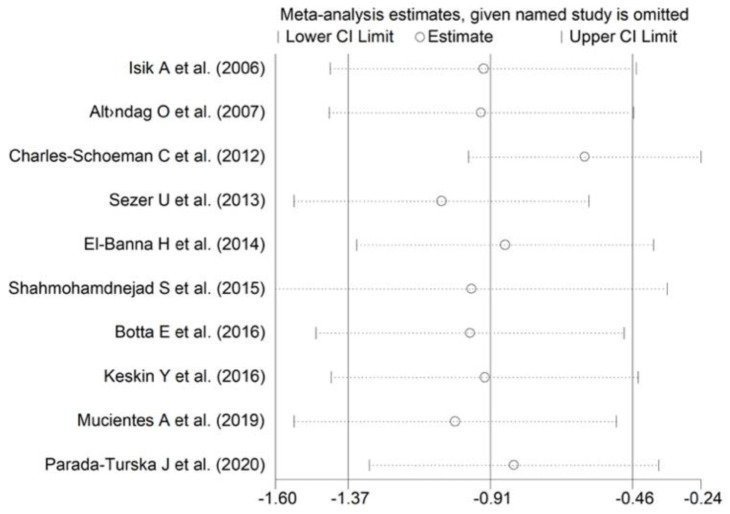
Sensitivity analysis of the association between ARE activity and RA. For each study, the displayed effect size (hollow circles) corresponds to the overall effect size computed from a meta-analysis excluding that study.

**Figure 9 antioxidants-11-02317-f009:**
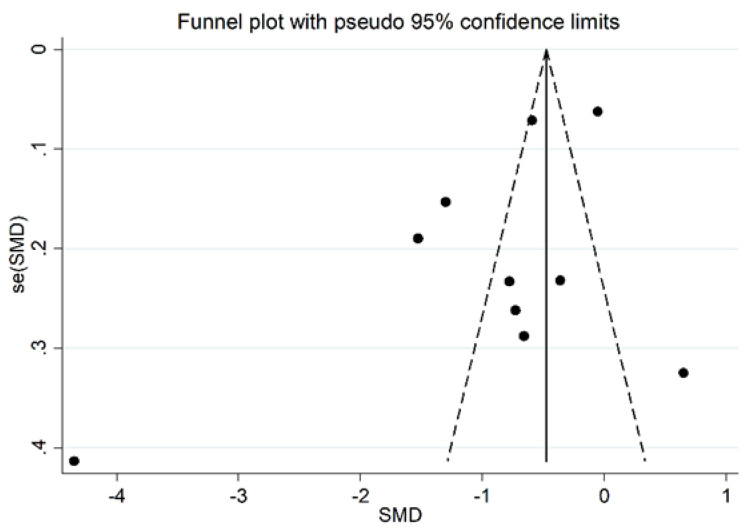
Funnel plot of the 10 retrieved studies evaluating the association between serum ARE concentration and RA disease.

**Figure 10 antioxidants-11-02317-f010:**
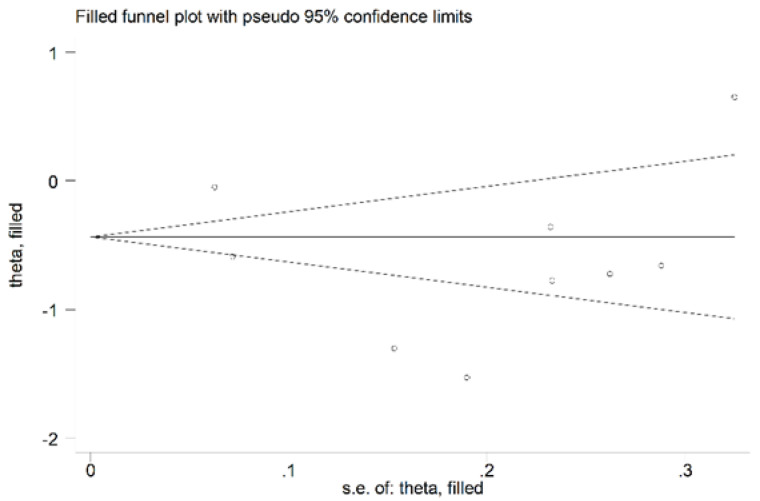
Funnel plot of studies investigating the association between serum ARE values and RA disease after trimming and filling. Dummy studies and genuine studies are represented by enclosed circles and free circles, respectively.

**Figure 11 antioxidants-11-02317-f011:**
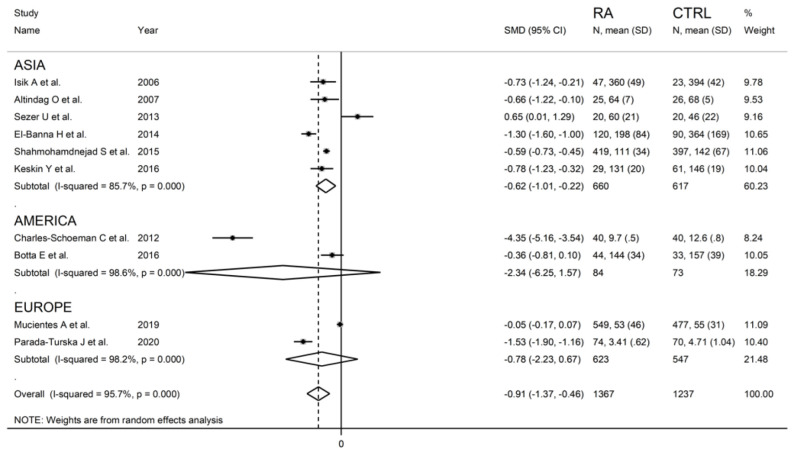
Forest plot of studies examining serum ARE values in RA patients and controls is shown according to the continent where the study was conducted.

**Table 1 antioxidants-11-02317-t001:** Summary of the studies in control and RA patients included in the meta-analysis.

	Controls					RA				
First Author, Year Country	n	Age (Years)	Gender (M/F)	PON-1 (Mean ± SD)	ARE (Mean ± SD)	n	Age (Years)	Gender (M/F)	PON-1 (Mean ± SD)	ARE (Mean ± SD)
Tanimoto N, 2003 Japan [29]	25	61	2/23	169 ± 67	--	25	61	2/23	125 ± 61	--
Baskol G, 2005 Turkey [30]	25	43	NR	225 ± 73	--	57	46	NR	168 ± 72	--
Altindag O, 2006 Turkey [31]	26	37	16/10	101 ± 44	68 ± 5	25	38	17/8	74 ± 17	64 ± 7
Isik A, 2006 Turkey [32]	23	50	6/17	149 ± 21	394 ± 42	47	50	7/40	84 ± 53	360 ± 49
Charles-Schoeman C, 2012 USA [33]	40	52	8/32	--	12.6 ± 0.8	40	56	6/34	--	9.7 ± 0.5
Sezer U, 2013, Turkey [34]	20	41	6/14	42 ± 52	46 ± 22	20	44	2/18	71 ± 51	60 ± 21
El-Banna H, 2014 Saudi Arabia [35]	90	41	30/60	152 ± 18	364 ± 169	120	41	30/90	64 ± 11	198 ± 84
Shahmohamadnejad S, 2015 Iran [36]	397	49	36/361	--	142 ± 67	419	49	42/377	--	111 ± 34
Bindal UD, 2016 India [37]	40	48	20/20	229 ± 17	--	40	49	20/20	156 ± 19	--
Botta, 2016 Argentina [38]	33	49	7/26	128 ± 45	157 ± 39	44	56	5/39	115 ± 38	144 ± 34
Keskin Y, 2016 Turkey [39]	61	41	20/41	84 ± 53	146 ± 19	29	39	9/20	70 ± 51	131 ± 20
Rodriguez-Carrio J, 2016 Spain [40]	110	54	29/81	312 ± 99	--	212	58	37/175	287 ± 236	--
O’Neill F, 2018 UK [41]	18	56	8/10	161 ± 53	--	18	59	6/12	143 ± 20	--
Shevchuk SV, 2018 Ukraine [42]	25	NR	NR	138 ± 19	--	67	NR	18/49	107 ± 24	--
Yang L, 2018 China [43]	12	43	3/9	440 ± 60	--	24	47	6/18	409 ± 39	--
Mucientes A, 2019 Spain [44]	477	NR	234/243	--	55 ± 31	549	NR	170/379	--	53 ± 46
Atwa ET, 2020 Egypt [45]	51	40	NR	20 ± 14	--	48	42	NR	4.6 ± 4.7	--
Bassu S, 2020 Italy [46]	101	55	50/51	128 ± 76	--	164	55	62/102	110 ± 67	--
Parada-Turska J, 2020 Poland [47]	70	52	12/58	6.79 ± 0.69	55 ± 31	74	55	14/60	4.88 ± 0.69	53 ± 46
Alisik T, 2021 Turkey [48]	67	52	7/60	228 ± 106	--	130	55	23/107	195 ± 131	--

ARE: arylesterase; NR: Not Reported; PON = Paraoxonase.

## Data Availability

All data relevant to the study are included in the article.

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
