# Peer review of "Association between Paraoxonase/Arylesterase Activity of Serum PON-1 Enzyme and Rheumatoid Arthritis: A Systematic Review and Meta-Analysis"

_antioxidants, 2022, doi:10.3390/antiox11122317_

Round 1
Reviewer 1 Report
In the paper, the authors have performed a meta-analysis of studies evaluating the activity of serum PON-1 and ARE in rheumatoid arthritis (RA) patients. The study complied with the PRISMA statements and was registered in PROSPERO (CRD42022345380). A random-effect model of meta-analysis was performed and the risk of bias with the Joanna Briggs Institute Critical-Appraisal Checklist and the certainty of evidence using GRADER was calculated. Overall, seventeen studies were found for PON-1 activity (1,144 RA patients, 797 controls) and 10 for ARE activity (1,367 RA patients, 1,037 controls). In conclusion, both serum PON-1 and ARE activity are significantly lower in RA patients.
Overall, the paper is well-designed and structured. The authors have used appropriate statistical methods for the meta-analysis. A critical analysis of the results has been performed and detailed throughout the paper and in the Discussion section.
We endorse the publication of the current paper as is, in Antioxidants
Author Response
We thank the reviewer for their careful reading of the manuscript and their constructive remarks
Reviewer 2 Report
The idea about relationship between oxidative stress, systemic inflammation and autoimmune response in RA is not new. However, this manuscript is a very nice study aimed to confirm with meta-analysis methodology the existence of reduced PON-1 and ARE activity in RA patients when compared to controls. The next orientation in the research is also formulated as a detailed population study is recommended for analysis of atherosclerosis and cardiovascular diseases occurrence in RA patients.
The manuscript need only small revision to correct a few errors in text.
Please correct:
line 30, ten instead of 10
line 37, disease instead of group
line 64 and 65, , instead of ;
further please check once more the complete text.....
It would be also nice to have some scientific opinion/explanation about the article No. 34 in which the levels of PON-1 and ARE where higher as in healthy controls. Further also the different behaviour between Asia, Africa and Europe population could be discussed, mainly for ARE.
Author Response
Correction have been made as suggested and the following sentence has been added to the text to discuss the issue of difference in PON-1 and ARE activity across studies.
“In one study [34] PON-1 and ARE activity were higher in RA than controls. The reason behind this discrepancy is not clear, but difference in the laboratory methodology and in the specific characteristics of population enrolled in the study (e.g., inflammatory state, cholesterol profile, and drugs) may have affected these results”
Reviewer 3 Report
Erre et al provided a detailed survey of research papers regarding paraoxonase activity and ARE activity for PON-1 enzyme and showed that both activities are lower in RA patients. The result summarized by the authors are convincing to lead to their conclusions. Some points: It would be better if the authors can give some details about the PON-1, ARE, and lactonase activity, eg what kind of enzymatic reactions are performed. How much are the molecular mechanism understood for PON-1? Since the structure of PON-1 has been solved before, are the three types of activity reported or predicted to come from the same domain or different domains/pockets? Apparently, both PON-1 activity and ARE activity are lower in RA patients, when directly comparing the two activities from different papers, are they correlated? More detailed figure captions would be helpful for the readers.
Author Response
1) As suggested by the reviewers we added some details about the PON-1, ARE, and lactonase activity in the following sentences in the text:
"PON-1 is a 43 kDa calcium-dependent glycoprotein with 355 amino acid residues [49,50]. It is in the liver and released into the circulation where it was associated mainly in HDLs and, to a lesser extent, in very low-density lipoproteins and chylomicrons [51]. HDL-bound PON-1 shows higher enzymatic activity than free PON-1. PON-1 is then transferred from the liver to several tissues [52] in which binding to cell membranes act protecting lipids against peroxidation [51]. In addition, PON1 protect low-density lipoprotein (LDL) from oxidation and have important anti- inflammatory effects [53]. Another important role of PON-1 is to protect against exposure of specific organophosphate (OP), by arylesterase activity. Studies on knockout mice have shown that PON-1 provide a significant protection against exposure to the OPs chlorpyrifos oxon and diazoxon [54]. Furthermore, PON-1 shows a lactonase activity that allows to hydrolyze homocysteine thiolactone (HCTL). HCTL is a toxic metabolite that reacts with protein lysines, thus leading protein inactivation and dysfunction. It is amply described as increased HCTL serum concentrations are associated with an increased risk of developing cardiovascular, neurological and autoimmune diseases [55–58]. However, the physiological relevance of PON-1 regarding HCTL is uncertain due to its low specific activity [59]. Therefore, PON-1 exhibit a broad range of enzymatic activities towards various types of substrates (lactonase, thiolactonase, arylesterase and aryldialkylphosphatase activities) in a single active site [60].
2) We tried to improve figure captions as suggested